# An Application of Optimal Control to Sugarcane Harvesting in Thailand

**Wisanlaya Pornprakun** [1,2,†] **, Surattana Sungnul** [1,2,*,†] **, Chanakarn Kiataramkul** [1,2,†]
**and Elvin James Moore** [1,2,†]

1   Department of Mathematics, King Mongkut's University of Technology North Bangkok, Pracharat 1 Road, Bangkok 10800, Thailand; wisanlaya5140662031@hotmail.com (W.P.); chanakarn.k@sci.kmutnb.ac.th (C.K.); elvin.j@sci.kmutnb.ac.th (E.J.M.)
2   Centre of Excellence in Mathematics, CHE, Si Ayutthaya Road, Bangkok 10400, Thailand
*   Correspondence: surattana.s@sci.kmutnb.ac.th
†   These authors contributed equally to this work.

**Abstract:** The sugar industry is of great importance to the Thai economy. In general, the government sets sugarcane prices at the beginning of each harvesting season based on type (fresh or fired), sweetness (sugar content) and gross weight. The main aim of the present research is to use optimal control to find optimal sugarcane harvesting policies for fresh and fired sugarcane for the four sugarcane producing regions of Thailand, namely North, Central, East and North-east, for harvesting seasons 2012/13, 2013/14, 2014/15, 2017/18 and 2018/19. The optimality problem is to determine the harvesting policy which gives maximum profit to the farmers subject to constraints on the maximum amount that can be cut in each day, where a harvesting policy is defined as the amount of each type of sugarcane harvested and delivered to the sugar factories during each day of a harvesting season. The results from the optimal control methods are also compared with results from three optimization methods, namely bi-objective, linear programming and quasi-Newton. The results suggest that discrete optimal control is the most effective of the five methods considered. The data used in this paper were obtained from the Ministry of Industry and the Ministry of Agriculture and Co-operatives of the Royal Thai government.

**Keywords:** discrete optimal control; continuous optimal control; bi-objective; linear programming; quasi-Newton methods; optimal sugarcane harvesting



## 1. Introduction

The main aims of this paper are: (1) to use two optimal control methods to determine optimal harvesting policies for sugarcane in Thailand, where an optimal harvesting policy is defined as the amount of sugarcane harvested each day during a crop year that gives maximum profit to farmers; (2) To compare the efficiency and results of the two optimal control methods with three optimization methods. The two optimal control methods we consider are based on the discrete and continuous Pontryagin maximum principles (see, e.g., [1–3]). The three optimization methods we consider are the bi-objective $\varepsilon$-constraints method (see, e.g., [4]), a quasi-Newton optimization method (see, e.g., [5–7]) and linear programming (see, e.g., [8]). We use the five optimal control and optimization methods to determine optimal harvesting policies for the two types of sugarcane (fresh and fired) in the four regions (North, Central, East and North-east) of Thailand for crop years 2012/13, 2013/14, 2014/15, 2017/18 and 2018/19. We use data obtained from the Ministry of Industry and the Ministry of Agriculture and Co-operatives of the Royal Thai government [9–13]. The present paper is an extension of a previous paper [14] in which bi-objective and quasi-Newton methods were used to find optimal harvesting policies for the crop years 2012/13, 2013/14 and 2014/15. For these two optimization methods, it was necessary to divide

a harvesting season into 15-day periods and to assume that there was zero growth of sugarcane during a season.

Although there have been few, if any, applications of optimal control to sugarcane harvesting, there have been applications of optimal control to similar types of problems in other areas which could be adapted to sugarcane farming [3]. For example, in 2011, Kiataramkul et al. [15] studied an optimal control problem for optimal nutritional intake for fetal growth in sheep. In 2014, Dine, Lenhart and Behncke [16] investigated discrete-time optimal harvesting of fish populations with age structure with the two objectives of maximizing the profit of fishing by finding the optimal harvesting strategy for each age class and of finding the optimal net size. In 2015, Kiataramkul and Matkhao [17] studied the optimal control problem of food intake of swine during the post weaning period with the objective of minimizing the amount of food fed to the swine and achieving an optimal weight at time of sale. In 2016, Puengpo et al. [18] applied continuous and discrete optimal control models to find optimal methods for sheep and swine feeding problems.

In contrast to optimal control, there have been a number of applications of optimization methods to sugar production. Some examples of applications of optimization methods to sugarcane harvesting and sugar production include the following. In 2012, Gomes [19] studied a bi-objective mathematical model for choosing sugarcane varieties in Brazil which have a harvest biomass residual that could be used in electricity generation. The bi-objective optimization model aimed to minimize the cost of harvesting the residual biomass and to maximize the revenue from sale of the generated electricity. In 2016, Sungnul et al. [20] studied a bi-objective optimization model to find an optimal time of harvesting for sugarcane growers in the North-east region of Thailand. The aim of this work was to help farmers to find the optimal harvesting time in order to maximize the revenue and to minimize the cost. Sungnul et al. used the $\varepsilon$-constraints method [4] to find the optimal cutting pattern by using the revenue as the objective function and the costs as constraints. In 2017, Sungnul et al. [21] extended the work in [20] to find the optimal harvesting times for all of the four regions of Thailand. Quasi-Newton optimization methods are well-known methods of optimization that have been used for many years to find optimal solutions for problems in many areas of science, finance and industry (see, e.g., [5–7]). In 2019, Pornprakun et al. [14] used the bi-objective and quasi-Newton optimization methods (see, e.g., [5–7]) to determine optimal harvesting policies for sugarcane in Thailand in order to maximize the total revenue and minimize harvesting cost for crop years 2012/13, 2013/14 and 2014/15.

## 2. Mathematical Models

We used the following mathematical model as the basic model for maximizing profit from sugarcane harvesting.

$$
\begin{aligned}
&\max_u J[u] = \int_{t_0}^{t_f} P(t, x(t), u(t))dt, \\
&\text{subject to } \frac{dx(t)}{dt} = rx(t)\left(1 - \frac{x(t)}{K}\right) - u(t), \\
&0 \le u(t) \le u_{max}, \ x(t) \ge 0, \ x(t_0) = a, \ x(t_f) = 0,
\end{aligned}
\tag{1}
$$

where $J[u]$ is the profit functional to be maximized by selecting the control variable $u(t)$, which is the rate (tonnes/day) of cutting sugarcane at time $t$. Also, $P(t, x(t), u(t))$ is the profit for sugarcane cut at time $t$ (baht/day),

　$P(t, x(t), u(t))$ is the profit for sugarcane cut at time $t$ (baht/day),
　$x(t)$ is the total amount of sugarcane (tonnes) on the farms in a region at time $t$,
　$u_{max}$ is the maximum rate of cutting sugarcane (tonnes/day),
　$t_0$ is the initial time at start of harvesting season (day),
　$t_f$ is the final time at end of harvesting season (day),
　$a$ is the total amount of sugarcane on farms at the initial time $t_0$ (tonnes),

$rx(t)\left(1 - \dfrac{x(t)}{K}\right)$ is a logistic growth function for the rate of increase in weight of sugarcane (tonnes per day) on the farms in a region at time $t$, where $r$ is a specific growth rate (1/day), and $K$ is a constant which represents the carrying capacity of the farms in the region (tonnes).

We also considered the following discrete version of (1).

$$
\max_u J[u] = \sum_{k=0}^{N-1} P(t_k, x_k, u_k),
$$

$$
\text{subject to } x_{k+1} = x_k + rx_k\left(1 - \frac{x_k}{K}\right) - u_k,
$$

$$
0 \le u_k \le u_{max}, \quad x_k \ge 0, \quad x_0 = x(t_0) = a, \quad x_N = x(t_N) = 0, \tag{2}
$$

where $x_k = x(t_k)$ and $u_k = u(t_k)$ and $t_k$, $k = 0, 1, 2, \ldots, N-1$ are $N$ periods of a cutting season and $t_N$ is the end of the cutting season. For the optimal control methods, we used 1-day periods, for the bi-objective and quasi-Newton methods we found it necessary to assume 15-day periods, and for linear programming we considered both 1-day and 15-day periods.

## 3. Optimization and Optimal Control Methods

In this section, we give details of the three optimization and two optimal control methods that we used to obtain optimal policies for sugarcane harvesting in Thailand.

### 3.1. Bi-Objective Optimization

For the bi-objective optimization problem, we separated the profit $P(t_k, x_k, u_k)$ per 15-day period into a revenue term $R(t_k, x_k, u_k)$ and a cost term $C(t_k, x_k, u_k)$ and assumed that $r = 0$, that is, no growth during the harvesting season. We then used the $\varepsilon$-constraints method [4,14] to solve the bi-objective optimization problem

$$
\max_u R(u) = \sum_{k=0}^{N-1} R(t_k, x_k, u_k),
$$

$$
\text{subject to } C(u) = \sum_{k=0}^{N-1} C(t_k, u_k, x_k) \le \varepsilon_r, \tag{3}
$$

$$
x_{k+1} = x_k - u_k,
$$

$$
0 \le u_k \le u_{max}, \quad x_k \ge 0, \quad x_0 = a, \quad x_N = 0,
$$

where the $\varepsilon_r$ are a set of values of costs between a minimum and a maximum value of the cost for a feasible cutting pattern.

### 3.2. Quasi-Newton Optimization

As in the previous paper [14], we used the constrained optimization function *fmincon* in Matlab with the "active-set" algorithm based on quasi-Newton method [5–7] to find the optimal cutting patterns to maximize the profit in (2) for 15-day periods and we assumed that there was no growth during the harvesting season.

### 3.3. Linear Programming

We used the Matlab program *linprog* with the "dual-simplex" algorithm based on linear programming [8] to solve the optimization problem in (2) for both 15-day periods and daily periods. We again assumed that there was no growth during the harvesting season because growth would make the optimization problem a nonlinear problem.

The results of three optimization methods for the years 2012/13, 2013/14 and 2014/15 have been published in [14] and the results for 2017/18, 2018/19 are included in this paper.

### 3.4. Pontryagin Maximum Principle for Continuous Optimal Control

For a given type, region and harvesting season, the continuous optimal control problem is of the form given in (1). We used the continuous Pontryagin maximum principle to solve this continuous optimal control problem (see, e.g., [1–3]). It should be noted that the optimal control problem for sugar harvesting has state variable constraints on the amount that can be cut each day and on the total amount that is available for cutting.

The first step is to define a Hamiltonian

$$H(t, x(t), u(t), \lambda(t)) = P(t, x(t), u(t)) + \lambda(t)^T f(t, x(t), u(t)), \tag{4}$$

where $f(t, x(t), u(t)) = rx(t)\left(1 - \frac{x(t)}{K}\right) - u(t)$ and the $\lambda(t)$ are called costate variables. Then, the equations for the state variables $x(t)$ and costate variables $\lambda(t)$ are given by

$$\text{State: } \frac{dx}{dt} = f(t, x(t), u(t)) = \frac{\partial H(t, x(t), u(t), \lambda(t))}{\partial \lambda(t)}, \tag{5}$$

$$
\begin{aligned}
\text{Costate: } \frac{d\lambda}{dt} &= -\frac{\partial H(t, x(t), u(t), \lambda(t))}{\partial x(t)} \\
&= -\frac{\partial P(t, x(t), u(t))}{\partial x(t)} - \lambda(t)^T \frac{\partial f(t, x(t), u(t))}{\partial x(t)}.
\end{aligned}
\tag{6}
$$

Since the boundary conditions on the state variables in (1) are given values at fixed initial and final times, the appropriate boundary conditions on the state and costate equations are

$$\text{State: } x(t_0) = a, \ x(t_f) = 0 \ \text{given.} \quad \text{Costate: } \lambda(0), \ \lambda(t_f) \ \text{free.} \tag{7}$$

Then, the Pontryagin maximum principle states that the optimal control $u^*(t)$ is obtained by finding the maximum of $H(t, x^*(t), u(t), \lambda^*(t))$ with respect to $u(t)$, where the $x^*(t)$, $\lambda^*(t)$ are the solutions of the state and costate equations for $u^*(t)$.

In general, the maximum of the Hamiltonian can occur at the minimum or maximum values of the cutting constraints or at an internal point. For the problem in (1) the state equation is a linear function of the control $u$. In this case, there is no internal optimal value of $u$ and the control is called a "bang-bang" control and the optimal values $u^*(t)$ are either the minimum $u(t) = 0$ or maximum $u(t) = u_{\max}$. However, as noted above, the sugar harvesting problem has the extra complication that the constraints on the control $u^*(t)$ also depend on the values of the state variables $x^*(t)$ because of the conditions $x^*(t) \geq 0$, $x^*(t_f) = 0$. We have included these conditions as explained in step 4 of the following algorithm.

Continuous optimal control algorithm

The algorithm that we used for the solution of the continuous optimal control problem is as follows.

1.  Select an initial feasible cutting pattern $u(t_k)$, $k = 0, 1, \ldots, N-1$ such that $x(t) \geq 0$ and $x(t_f) = 0$.
2.  Solve the state and costate equations for this pattern to obtain values of $x(t_k)$ and $\lambda(t_k)$. For reasons of numerical stability, the state equations must be integrated forwards in time and the costate equations must be integrated backwards in time.
3.  Compute the Hamiltonian values $H(t_k, x(t_k), u(t_k), \lambda(t_k))$ and the partial derivative values $\dfrac{\partial H(t_k, x(t_k), u(t_k), \lambda(t_k))}{\partial u(t_k)}$.
4.  In our examples, we have found that all partial derivatives are positive. Therefore, if there were no constraints on the state variables, the optimal cutting pattern would be to cut the maximum amount available for cutting each day. In order to satisfy the

state variable constraints $x(t) \geq 0$ and $x(t_f) = 0$, we chose the optimal $u(t_k)$ values as follows.

(a)    Sort the derivatives $\dfrac{\partial H(t_k, x(t_k), u(t_k), \lambda(t_k))}{\partial u(t_k)}$ in decreasing order.

(b)    Find the number of days $M$ required to cut all available cane at the maximum rate $u_{\max}$ and select the times $t_k$ corresponding to the top $M$ values of the derivatives.

(c)    If $Mu_{\max} = a$, set the new cutting pattern with $u(t_k) = u_{\max}$ at these $M$ values of $t_k$ and $u(t_k) = 0$ at all other times. However, if $Mu_{\max} > a$, set $u(t_k) = u_{\max}$ for the top $M - 1$ derivative values and
$u(t_k) = a - (M-1)u_{\max}$ where $t_k$ is time of derivative $M$.
Note: A slight modification is required if growth can occur ($r > 0$).

5.    Then, stop if the new cutting pattern is within a selected tolerance of the old cutting pattern or return to step 2 and repeat if the new pattern is not within the selected tolerance of the old.

### 3.5. Pontryagin Maximum Principle for Discrete Optimal Control

The discrete version of the continuous optimal control problem in (1) is as follows.

$$
\begin{aligned}
\max_u J[u] &= \sum_{k=0}^{N-1} P(t_k, x_k, u_k), \\
\text{subject to } x_{k+1} &= x_k + rx_k\left(1 - \frac{x_k}{K}\right) - u_k, \\
0 &\leq u_k \leq u_{max}, \quad x_0 = x(t_0) = a, \quad x_N = x(t_N) = 0,
\end{aligned}
\tag{8}
$$

where $x_k = x(t_k)$ and $u_k = u(t_k)$.

As for the continuous case, the first step is to define a Hamiltonian

$$
H(k, x_k, u_k, \lambda_{k+1}) = P(t_k, x_k, u_k) + \lambda_{k+1}^T f(k, x_k, u_k),
\tag{9}
$$

where $f(k, x_k, u_k) = x_k + rx_k\left(1 - \frac{x_k}{K}\right) - u_k$.

Then, the equations for the state variables $x_k$ and costate variables $\lambda_k$ are given by

$$
\text{State: } x_{k+1} = f(k, x_k, u_k) = \frac{\partial H(k, x_k, u_k, \lambda_{k+1})}{\partial \lambda_{k+1}},
\tag{10}
$$

$$
\begin{aligned}
\text{Costate: } \lambda_k &= \frac{\partial H(k, x_k, u_k, \lambda_{k+1})}{\partial x_k} \\
&= \frac{\partial P(k, x_k, u_k)}{\partial x_k} + \lambda_{k+1}^T \frac{\partial f(k, x_k, u_k)}{\partial x_k}.
\end{aligned}
\tag{11}
$$

Since the boundary conditions on the state variables in (8) are given values at fixed initial and final times, the appropriate boundary conditions on the state and costate equations are

$$
\text{State: } x_0 = a, \ x_N = 0 \ \text{given.} \qquad \text{Costate: } \lambda_0, \ \lambda_N \ \text{free.}
\tag{12}
$$

As for the continuous case, the optimal cutting pattern will be "bang-bang". The algorithm for finding the optimal cutting pattern for the continuous case can also be used for the discrete case.

## 4. The Commercial Cane Sugar System (CCS) in Thailand

As details of the CCS system have been given in [14,22], we will only give a brief review here. Thai farmers harvest two types of sugarcane, namely fresh and fired, where

fired sugarcane is burnt to remove leaves before it is cut so that it can be cut manually by workers, whereas fresh sugarcane is usually cut by machines which can remove the leaves as the cane is cut. In the CCS system, the Royal Thai government sets the price of fresh and fired sugarcane for each of the four sugarcane producing regions in Thailand, namely North, Central, East and North-east. The price of the sugarcane is based on the two main factors of weight and sweetness, where sweetness or CCS is the percentage of sugar in the sugarcane. All data in this paper have been obtained from the Office of the Cane and Sugar Board (OCSB), Ministry of Industry and the Ministry of Agriculture and Co-operatives of the Royal Thai government [9–13]. The OCSB reports data for 15-day periods of a harvesting season which typically starts around 15th November and ends around 15th June of the following year.

In the following, we will use the notation, $i = A$ for fresh and $i = B$ for fired sugarcane. For the four sugarcane-growing regions of Thailand, we will use the notation $j = 1$ for North, $j = 2$ for Central, $j = 3$ for East, and $j = 4$ for North-east.

### 4.1. Price of Sugarcane

As stated above, the price of sugarcane is set on the basis of type, weight, and sweetness (CCS) [14,20].

1. Price based on weight: The basic price per weight of sugarcane (baht/tonne) is fixed by the Royal Thai government for each crop year. This basic price is the same for all regions. However, the actual price paid to farmers for fired sugarcane is 30 baht/tonne less than the basic price. Then, at the end of harvesting for the year, factories in each region will share the total amount of money deducted from fired sugarcane sales in that region to farmers who sold fresh sugarcane at a rate not exceeding 70 baht/tonne of fresh sugarcane delivered. We will use the notation $P_w(i, j)$ to denote actual price based on weight (baht/tonne) for type $i$ in region $j$.

   The actual prices per tonne based on weight for sugarcane for a given crop year are:

$$\text{Fresh: } P_w(A, j) = P_w + \frac{30a(B, j)}{a(A, j)} \leq P_w + 70, \qquad \text{Fired: } P_w(B) = P_w - 30, \quad (13)$$

   where $P_w$ is the basic price, $a(A, j)$ is the total amount of fresh sugarcane (tonnes) harvested in region $j$ and $a(B, j)$ is the total amount of fired sugarcane (tonnes) harvested in region $j$ for a given year.

2. Price based on sweetness (CCS): Each year, the Royal Thai government sets a basic price per tonne for sugarcane with 10 CCS, where CCS is the percentage by weight of sugar in sugarcane. This price is the same for fresh and fired sugarcane and for all regions. The actual price per tonne received by farmers is then adjusted if the CCS is different from 10. For sugarcane harvested in period $k$ in region $j$ the actual price per tonne for a given year is

$$P_c(k, j) = P_c(1 + 0.06y(k, j)), \qquad (14)$$

   where $P_c$ is the basic price per tonne for sugarcane with 10 CCS set by the government for the year, and $y(k, j) = \text{CCS}(k, j) - 10$, where $\text{CCS}(k, j)$ is the average CCS from sugarcane harvested in period $k$ in region $j$ and the factor 0.06 is the rate of change of price per 1 CCS from the base level of 10.

   Therefore, the total prices to farmers (baht/tonne) for a given crop year are:

$$\text{Fresh: } P(k, A, j) = P_w(A, j) + P_c(k, j), \qquad \text{Fired: } P(k, B, j) = P_w(B) + P_c(k, j). \quad (15)$$

The basic prices per weight $P_w$ and for sweetness $P_c$ set by the Royal Thai government and the total amounts delivered to the factories are shown in Table 1. It can be seen from this data that the amount of fresh sugarcane is appreciably less than the amount of fired in

all regions in all crop years and that the amount of sugarcane was increasing from 2012/13 to 2017/18 and then decreased in 2018/19.

**Table 1.** The basic prices (Baht/tonne) set by the Royal Thai Government and the total amounts ($10^6$ tonnes) of fresh and fired sugarcane delivered to the factories in each region in each crop year.

| Type | Year | Basic Price (baht) | | Total Amount (tonnes) | | | |
|---|---|---|---|---|---|---|---|
| | | $P_w$ | $P_c$ | North | Central | East | North-East |
| **Fresh** | 2012/13 | 198.47 | 999.20 | 7.036 | 9.309 | 0.955 | 16.909 |
| | 2013/14 | 194.67 | 958.31 | 7.076 | 9.600 | 1.235 | 20.013 |
| | 2014/15 | 197.41 | 900.00 | 7.384 | 9.235 | 1.264 | 19.028 |
| | 2017/18 | 141.00 | 880.00 | 11.399 | 11.735 | 1.741 | 20.623 |
| | 2018/19 | 141.00 | 700.00 | 13.944 | 11.069 | 1.705 | 24.221 |
| **Fired** | 2012/13 | 140.00 | 999.20 | 17.562 | 21.189 | 3.731 | 23.311 |
| | 2013/14 | 140.00 | 958.31 | 17.134 | 20.479 | 3.237 | 24.892 |
| | 2014/15 | 140.00 | 900.00 | 17.999 | 18.766 | 3.946 | 28.338 |
| | 2017/18 | 140.00 | 880.00 | 21.641 | 24.858 | 4.942 | 37.989 |
| | 2018/19 | 140.00 | 700.00 | 19.163 | 20.468 | 4.174 | 36.226 |

The CCS values are shown in Tables 2 and 3 for 15-day periods for all regions for crop years 2017/18 and 2018/19. Similar tables for the crop years 2012/13, 2013/14 and 2014/15 have been published previously in [14]. It can be seen that in 2017/18 the CCS value in all regions reached a maximum in April and then decreased slightly, whereas in 2018/19 the CCS value increased in all regions during the harvesting season.

**Table 2.** The Commercial Cane Sugar System (CCS) value in each region for crop year 2017/18.

| Period | North | Central | East | North-East |
|---|---|---|---|---|
| (1) 1–15 December 2017 | 10.387 | 9.932 | 10.482 | 11.655 |
| (2) 16–31 December 2017 | 10.726 | 10.262 | 10.807 | 12.009 |
| (3) 1–15 January 2018 | 11.032 | 10.485 | 10.994 | 12.313 |
| (4) 16–31 January 2018 | 11.378 | 10.856 | 11.367 | 12.596 |
| (5) 1–14 February 2018 | 11.654 | 11.134 | 11.670 | 12.840 |
| (6) 15–28 February 2018 | 11.856 | 11.328 | 11.911 | 13.021 |
| (7) 1–15 March 2018 | 12.005 | 11.463 | 12.075 | 13.174 |
| (8) 16–31 March 2018 | 12.120 | 11.586 | 12.211 | 13.291 |
| (9) 1–15 April 2018 | 12.147 | 11.603 | 12.253 | 13.333 |
| (10) 16–30 April 2018 | 12.139 | 11.595 | 12.265 | 13.337 |
| (11) 1–15 May 2018 | 12.112 | 11.572 | 12.235 | 13.311 |
| (12) 16–31 May 2018 | 12.101 | 11.564 | 12.189 | 13.298 |
| (13) 1–15 June 2018 | 12.101 | 11.561 | 12.187 | 13.298 |

The total prices for the fresh and fired sugarcane are shown in Tables 4 and 5 for 15-day periods for all regions for crop years 2017/18 and 2018/19. It can be seen that the prices in all regions increase rapidly at the beginning of a crop year and then slowly at the end due to the changes in CCS values. It can also be seen that the fresh prices are appreciably higher than the fired prices due to the price adjustments discussed in Section 4.1.

**Table 3.** The Commercial Cane Sugar System (CCS) value in each region for crop year 2018/19.

| Period | North | Central | East | North-East |
|---|---|---|---|---|
| (1) 16–30 November 2018 | 9.760 | 9.752 | - | 11.530 |
| (2) 1–15 December 2018 | 10.175 | 10.114 | 11.316 | 11.909 |
| (3) 16–31 December 2018 | 10.486 | 10.322 | 11.641 | 12.163 |
| (4) 1–15 January 2019 | 10.822 | 10.622 | 11.855 | 12.507 |
| (5) 16–31 January 2019 | 11.182 | 10.961 | 12.176 | 12.819 |
| (6) 1–14 February 2019 | 11.444 | 11.213 | 12.404 | 13.036 |
| (7) 15–28 February 2019 | 11.639 | 11.419 | 12.599 | 13.179 |
| (8) 1–15 March 2019 | 11.834 | 11.596 | 12.790 | 13.311 |
| (10) 16–31 March 2019 | 11.986 | 11.654 | 12.858 | 13.427 |
| (11) 1–15 April 2019 | 12.015 | 11.658 | 12.858 | 13.464 |
| (12) 16–30 April 2019 | 12.015 | 11.658 | - | 13.481 |
| (13) 1–15 May 2019 | 12.015 | 11.658 | - | 13.480 |

Note: The-means that no sugarcane was delivered to the mills in that period.

**Table 4.** The price (baht/tonne) of fresh and fired sugarcane in each region for crop year 2017/18.

| Period | North | | Central | | East | | North-East | |
|---|---|---|---|---|---|---|---|---|
| | Fresh | Fired | Fresh | Fired | Fresh | Fired | Fresh | Fired |
| (1) 1–15 December 2017 | 1041.45 | 989.76 | 1017.42 | 966.54 | 1046.47 | 994.60 | 1108.38 | 1054.40 |
| (2) 16–31 December 2017 | 1059.32 | 1007.02 | 1034.82 | 983.35 | 1063.59 | 1011.14 | 1127.10 | 1072.48 |
| (3) 1–15 January 2018 | 1075.49 | 1022.63 | 1046.63 | 994.76 | 1073.47 | 1020.68 | 1143.12 | 1087.96 |
| (4) 16–31 January 2018 | 1093.77 | 1040.29 | 1066.19 | 1013.65 | 1093.16 | 1039.70 | 1158.09 | 1102.42 |
| (5) 1–14 February 2018 | 1108.35 | 1054.37 | 1080.86 | 1027.82 | 1109.15 | 1055.15 | 1170.95 | 1114.83 |
| (6) 15–28 February 2018 | 1118.99 | 1064.65 | 1091.12 | 1037.73 | 1121.90 | 1067.46 | 1180.48 | 1124.05 |
| (7) 1–15 March 2018 | 1126.88 | 1072.27 | 1098.26 | 1044.62 | 1130.55 | 1075.82 | 1188.59 | 1131.87 |
| (8) 16–31 March 2018 | 1132.94 | 1078.12 | 1104.74 | 1050.88 | 1137.74 | 1082.76 | 1194.75 | 1137.83 |
| (9) 1–15 April 2018 | 1134.36 | 1079.49 | 1105.62 | 1051.74 | 1139.97 | 1084.91 | 1197.00 | 1140.00 |
| (10) 16–30 April 2018 | 1133.94 | 1079.09 | 1105.23 | 1051.36 | 1140.58 | 1085.50 | 1197.20 | 1140.19 |
| (11) 1–15 May 2018 | 1132.52 | 1077.72 | 1104.02 | 1050.19 | 1139.02 | 1084.00 | 1195.82 | 1138.86 |
| (12) 16–31 May 2018 | 1131.96 | 1077.17 | 1103.56 | 1049.75 | 1136.59 | 1081.65 | 1195.14 | 1138.21 |
| (13) 1–15 June 2018 | - | - | 1103.43 | 1049.62 | 1136.46 | 1081.53 | 1195.14 | - |

Note: The-means that no sugarcane was delivered to the mills in that period.

**Table 5.** The price (baht/tonne) of fresh and fired sugarcane in each region for crop year 2018/19.

| Period | North | | Central | | East | | North-East | |
|---|---|---|---|---|---|---|---|---|
| | Fresh | Fired | Fresh | Fired | Fresh | Fired | Fresh | Fired |
| (1) 16–30 Nov 2018 | 830.94 | 780.37 | 830.57 | 780.02 | - | - | 905.28 | 851.52 |
| (2) 1–15 December 2018 | 848.33 | 797.02 | 845.77 | 794.57 | 896.29 | 842.92 | 921.17 | 866.73 |
| (3) 16–31 December 2018 | 861.41 | 809.54 | 854.53 | 802.95 | 909.92 | 855.97 | 931.83 | 876.94 |
| (4) 1–15 January 2019 | 875.51 | 823.03 | 867.13 | 815.01 | 918.92 | 864.58 | 946.27 | 890.76 |
| (5) 16–31 January 2019 | 890.66 | 837.53 | 881.37 | 828.64 | 932.40 | 877.48 | 959.39 | 903.32 |
| (6) 1–14 February 2019 | 901.66 | 848.06 | 891.93 | 838.75 | 941.96 | 886.63 | 968.51 | 912.05 |
| (7) 15–28 February 2019 | 909.83 | 855.88 | 900.58 | 847.03 | 950.17 | 894.49 | 974.52 | 917.80 |
| (8) 1–15 March 2019 | 918.04 | 863.74 | 908.04 | 854.16 | 958.17 | 902.15 | 980.06 | 923.10 |
| (10) 16–31 March 2019 | 924.40 | 869.82 | 910.46 | 856.48 | 961.05 | 904.91 | 984.95 | 927.78 |
| (11) 1–15 April 2019 | 925.61 | 870.99 | 910.66 | 856.67 | 961.05 | 904.90 | 986.50 | 929.26 |
| (12) 16–30 April 2019 | - | - | 910.66 | 856.67 | - | - | 987.18 | 929.92 |
| (13) 1–15 May 2019 | - | - | - | - | - | - | 987.18 | 929.91 |

Note: The-means that no sugarcane was delivered to the mills in that period.

*4.2. Costs of Production*

The costs of production (baht/tonne) can be separated into cutting costs, transport costs and maintenance costs. In this paper, we assume that the cutting and transport costs are fixed costs that depend only on the amount cut in a given period, whereas the

maintenance costs (baht/tonne) are variable costs that depend on the amount of uncut sugarcane remaining on the farm after cutting in previous periods.

We assume that the total cost of sugarcane production (baht) of type $i$ in period $k$ in region $j$ in a given year is

$$C(i,j) = \sum_{k=0}^{N-1} (C_h(i,j)u_k(i,j) + C_t(i,j)u_k(i,j) + C_m(i,j)x_k(i,j)), \tag{16}$$

where $u_k(i,j)$ is the weight of sugarcane (tonnes) cut in period $k$, $x_k(i,j)$ is the weight of uncut sugarcane remaining on the farm in period $k$, $C_h(i,j)$ is harvesting cost of sugarcane (baht/tonne), $C_t(i,j)$ is transport cost for delivering sugarcane to the mills (baht/tonne) and $C_m(i,j)$ is maintenance cost (baht/tonne) for sugarcane remaining on the farms.

The harvesting, transport and maintenance costs obtained from the OCSB are shown in Table 6 for each region for each crop year. It can be seen that the harvesting costs are lower in all regions for the years 2017/18 and 2018/19 for both fresh and fired cane, whereas there is no clear pattern for the transport and maintenance costs.

**Table 6.** The average costs of sugarcane production (baht/tonne) in each region for each crop year.

| Year | Region | Harvesting Cost $C_h(i,j)$ (baht/tonne) | Transport Cost $C_t(i,j)$ (baht/tonne) | Maintenance Cost $C_m(i,j)$ (baht/tonne) |
|---|---|---|---|---|
| 2012/13 | North | 924.28 | 136.51 | 52.70 |
| | Central | 872.95 | 147.66 | 94.24 |
| | East | 836.25 | 148.89 | 92.51 |
| | North-east | 765.18 | 141.97 | 56.30 |
| 2013/14 | North | 815.89 | 149.46 | 66.07 |
| | Central | 781.41 | 147.04 | 86.59 |
| | East | 912.61 | 165.66 | 106.01 |
| | North-east | 875.84 | 151.64 | 59.55 |
| 2014/15 | North | 1061.42 | 182.36 | 79.38 |
| | Central | 954.16 | 155.30 | 82.32 |
| | East | 1024.71 | 194.89 | 94.30 |
| | North-east | 987.00 | 140.12 | 91.71 |
| 2017/18 | North | 771.27 | 143.62 | 84.94 |
| | Central | 771.91 | 137.22 | 84.83 |
| | East | 848.51 | 146.21 | 102.43 |
| | North-east | 871.78 | 150.74 | 66.80 |
| 2018/19 | North | 738.86 | 138.40 | 104.87 |
| | Central | 754.86 | 138.24 | 83.26 |
| | East | 768.63 | 157.68 | 75.48 |
| | North-east | 775.09 | 149.58 | 59.69 |

*4.3. Total Revenue and Total Profit*

For each type $i$ and region $j$, the total revenue for a given year is

$$R(u(i,j)) = \sum_{k=0}^{N-1} P(k,i,j)u_k(i,j), \tag{17}$$

and the total profit is

$$J(u(i,j)) = \sum_{k=0}^{N-1} (P_t(k,i,j) - C_h(i,j) - C_t(i,j))u_k(i,j) - \sum_{k=0}^{N-1} C_m(i,j)x_k(i,j). \tag{18}$$

*4.4. Cutting Constraints*

We assume that there are constraints $u_{max}(i,j)$ on the amount of sugarcane of type $i$ that can be cut each day in a given region $j$. These constraints could be due, for example, to the number of machines available for cutting fresh sugarcane or the number of workers avaiable for cutting fired sugarcane. In addition, there can be constraints due to the number of trucks available for transporting the cut cane to the mill and the cutting capacities of the

mills. We also add the constraints that $0 \leq u_k(i,j) \leq x_k(i,j)$ and that all sugarcane must be cut, that is, $x_N(i,j) = 0$ at the end of the harvesting season.

## 5. Results

### 5.1. Profits

Examples of the optimal profits computed from the three optimization and two optimal control methods are shown in Table 7 for fresh sugarcane for the four regions of Thailand for the crop year 2017/18. It can be seen that all optimization and optimal control methods give the same results. A comparison of the theoretical results with the actual profit show that the best agreement is for mcf = 1, that is, full maintenance cost with an upper limit of approximately half of the total amount available cut each 15-day period. We have obtained similar results for fresh and fired sugarcane for the four regions of Thailand for the crop years 2012/13, 2013/14, 2014/15 and 2017/18.

**Table 7.** The total profit ($\times 10^{10}$ Baht) of fresh sugarcane in each region in crop year 2017/18.

| Region | mcf | P.O.C. | Actual Profit | Total Profit ($10^{10}$ Baht) | | | | |
|---|---|---|---|---|---|---|---|---|
| | | | | Bi-obj | Lin prog. | qn | Disc. | Cont. |
| North | 0 | 0.2 | | 0.92808 | 0.92808 | 0.92808 | 0.92806 | 0.92806 |
| | | 0.4 | | 0.92906 | 0.92906 | 0.92906 | 0.92906 | 0.92906 |
| | | 0.6 | | 0.92929 | 0.92929 | 0.92929 | 0.92932 | 0.92932 |
| | | 0.8 | | 0.92935 | 0.92935 | 0.92935 | 0.92939 | 0.92939 |
| | | 1.0 | | 0.92940 | 0.92940 | 0.92940 | 0.92942 | 0.92942 |
| | 0.25 | 0.2 | 0.11776 | 0.38605 | 0.38609 | 0.38609 | 0.38598 | 0.38598 |
| | | 0.4 | | 0.59645 | 0.59662 | 0.59662 | 0.60516 | 0.60516 |
| | | 0.6 | | 0.66648 | 0.66657 | 0.66657 | 0.67816 | 0.67816 |
| | | 0.8 | | 0.70163 | 0.70167 | 0.70167 | 0.71462 | 0.71462 |
| | | 1.0 | | 0.73677 | 0.73677 | 0.73677 | 0.73652 | 0.73652 |
| | 1 | 0.2 | | −1.06910 | −1.06910 | −1.06910 | −1.06922 | −1.06920 |
| | | 0.4 | | −0.16008 | −0.16008 | −0.16008 | −0.12257 | −0.12257 |
| | | 0.6 | | 0.14270 | 0.14270 | 0.14270 | 0.19309 | 0.19309 |
| | | 0.8 | | 0.29421 | 0.29421 | 0.29421 | 0.35062 | 0.35062 |
| | | 1.0 | | 0.44573 | 0.44573 | 0.44573 | 0.44548 | 0.44548 |
| Central | 0 | 0.2 | | 0.93354 | 0.93354 | 0.93353 | 0.93353 | 0.93353 |
| | | 0.4 | | 0.93447 | 0.93447 | 0.93445 | 0.93445 | 0.93445 |
| | | 0.6 | | 0.93471 | 0.93471 | 0.93466 | 0.93466 | 0.93466 |
| | | 0.8 | | 0.93475 | 0.93475 | 0.93472 | 0.93472 | 0.93472 |
| | | 1.0 | | 0.93480 | 0.93480 | 0.93475 | 0.93475 | 0.93475 |
| | 0.25 | 0.2 | 0.09694 | 0.37482 | 0.37482 | 0.37463 | 0.37463 | 0.37463 |
| | | 0.4 | | 0.59408 | 0.59418 | 0.60280 | 0.60280 | 0.60280 |
| | | 0.6 | | 0.66786 | 0.66802 | 0.68011 | 0.68011 | 0.68011 |
| | | 0.8 | | 0.70506 | 0.70535 | 0.71878 | 0.71878 | 0.71878 |
| | | 1.0 | | 0.74269 | 0.74269 | 0.74186 | 0.74186 | 0.74186 |
| | 1 | 0.2 | | −1.13210 | −1.13210 | −1.13228 | −1.13228 | −1.13230 |
| | | 0.4 | | −0.18941 | −0.18941 | −0.15079 | −0.15079 | −0.15079 |
| | | 0.6 | | 0.12553 | 0.12553 | 0.17780 | 0.17780 | 0.17780 |
| | | 0.8 | | 0.28342 | 0.28342 | 0.34185 | 0.34185 | 0.34185 |
| | | 1.0 | | 0.44131 | 0.44131 | 0.44048 | 0.44048 | 0.44048 |

**Table 7.** *Cont.*

| Region | mcf | P.O.C. | Actual Profit | Total Profit ($10^{10}$ Baht) | | | | |
|---|---|---|---|---|---|---|---|---|
| | | | | Bi-obj | Lin prog. | qn | Disc. | Cont. |
| East | 0 | 0.2 | | 0.14925 | 0.14925 | 0.14925 | 0.14924 | 0.14924 |
| | | 0.4 | | 0.14946 | 0.14946 | 0.14946 | 0.14946 | 0.14946 |
| | | 0.6 | | 0.14950 | 0.14950 | 0.14950 | 0.14951 | 0.14951 |
| | | 0.8 | | 0.14951 | 0.14951 | 0.14951 | 0.14952 | 0.14952 |
| | | 1.0 | | 0.14952 | 0.14952 | 0.14952 | 0.14953 | 0.14953 |
| | 0.25 | 0.2 | 0.00056 | 0.05108 | 0.05108 | 0.05108 | 0.05105 | 0.05105 |
| | | 0.4 | | 0.09044 | 0.09046 | 0.09046 | 0.09201 | 0.09201 |
| | | 0.6 | | 0.10376 | 0.10379 | 0.10379 | 0.10596 | 0.10596 |
| | | 0.8 | | 0.11047 | 0.11052 | 0.11052 | 0.11294 | 0.11294 |
| | | 1.0 | | 0.11725 | 0.11725 | 0.11725 | 0.11710 | 0.11710 |
| | 1 | 0.2 | | −0.21515 | −0.21515 | −0.21515 | −0.21518 | −0.21518 |
| | | 0.4 | | −0.04798 | −0.04798 | −0.04798 | −0.04113 | −0.04113 |
| | | 0.6 | | 0.00794 | 0.00794 | 0.00794 | 0.01722 | 0.01722 |
| | | 0.8 | | 0.03598 | 0.03598 | 0.03598 | 0.04635 | 0.04635 |
| | | 1.0 | | 0.06401 | 0.06401 | 0.06401 | 0.06386 | 0.06386 |
| North-east | 0 | 0.2 | | 1.81221 | 1.81220 | 1.81220 | 1.81210 | 1.81210 |
| | | 0.4 | | 1.81360 | 1.81360 | 1.81360 | 1.81370 | 1.81370 |
| | | 0.6 | | 1.81406 | 1.81410 | 1.81410 | 1.81420 | 1.81420 |
| | | 0.8 | | 1.81435 | 1.81440 | 1.81440 | 1.81440 | 1.81440 |
| | | 1.0 | | 1.81464 | 1.81460 | 1.81460 | 1.81450 | 1.81450 |
| | 0.25 | 0.2 | 0.14416 | 0.71914 | 0.71913 | 0.71913 | 0.71892 | 0.71892 |
| | | 0.4 | | 1.16030 | 1.16050 | 1.16050 | 1.17855 | 1.17850 |
| | | 0.6 | | 1.30590 | 1.30620 | 1.30620 | 1.33030 | 1.33030 |
| | | 0.8 | | 1.37813 | 1.37870 | 1.37870 | 1.40566 | 1.40570 |
| | | 1.0 | | 1.45124 | 1.45120 | 1.45120 | 1.45090 | 1.45090 |
| | 1 | 0.2 | | −2.26570 | −2.26570 | −2.26570 | −2.26595 | −2.26600 |
| | | 0.4 | | −0.39165 | −0.39165 | −0.39165 | −0.31416 | −0.31416 |
| | | 0.6 | | 0.23166 | 0.23166 | 0.23166 | 0.33534 | 0.33534 |
| | | 0.8 | | 0.54296 | 0.54296 | 0.54296 | 0.65904 | 0.65904 |
| | | 1.0 | | 0.85427 | 0.85427 | 0.85427 | 0.85393 | 0.85393 |

Note: P.O.C. means the percentage of total available sugarcane that can be cut in a 15-day period.

### 5.2. Optimal Daily Cutting Patterns

In this section, we show the optimal daily cutting patterns that we obtained from discrete optimal control. We found that continuous optimal control gave the same optimal cutting patterns. Also, for the special case of zero growth, we found that linear programming gave the same optimal daily cutting patterns. For comparison, optimal cutting patterns computed from the bi-objective and quasi-Newton methods have been published in [14] for 15-day cutting periods for the crop years 2012/13, 2013/14 and 2014/15.

The results in Figures 1 and 2 show the optimal cutting patterns for the crop year 2017/18 if prices, costs, and profits and an upper bound on the amount cut per day are the main factors involved in determining the optimal profits of the farmers. We have computed optimal daily cutting patterns for the crop years 2012/13, 2013/14, 2014/15, 2018/19 and obtained similar results. For each type, region and year, we examined the effect of changing the values of the upper bound on the maximum cutting per day $u_{max}$ (tonnes per day) and the effect of reducing the maintenance costs by a factor mcf ($0 \le$ mcf $\le 1$) of the actual maintenance cost. It can be seen that the optimal cutting patterns are sensitive to changes in the maintenance cost as they range from cutting all sugarcane as early as possible for mcf = 1 to cutting all sugarcane as late as possible for mcf = 0.

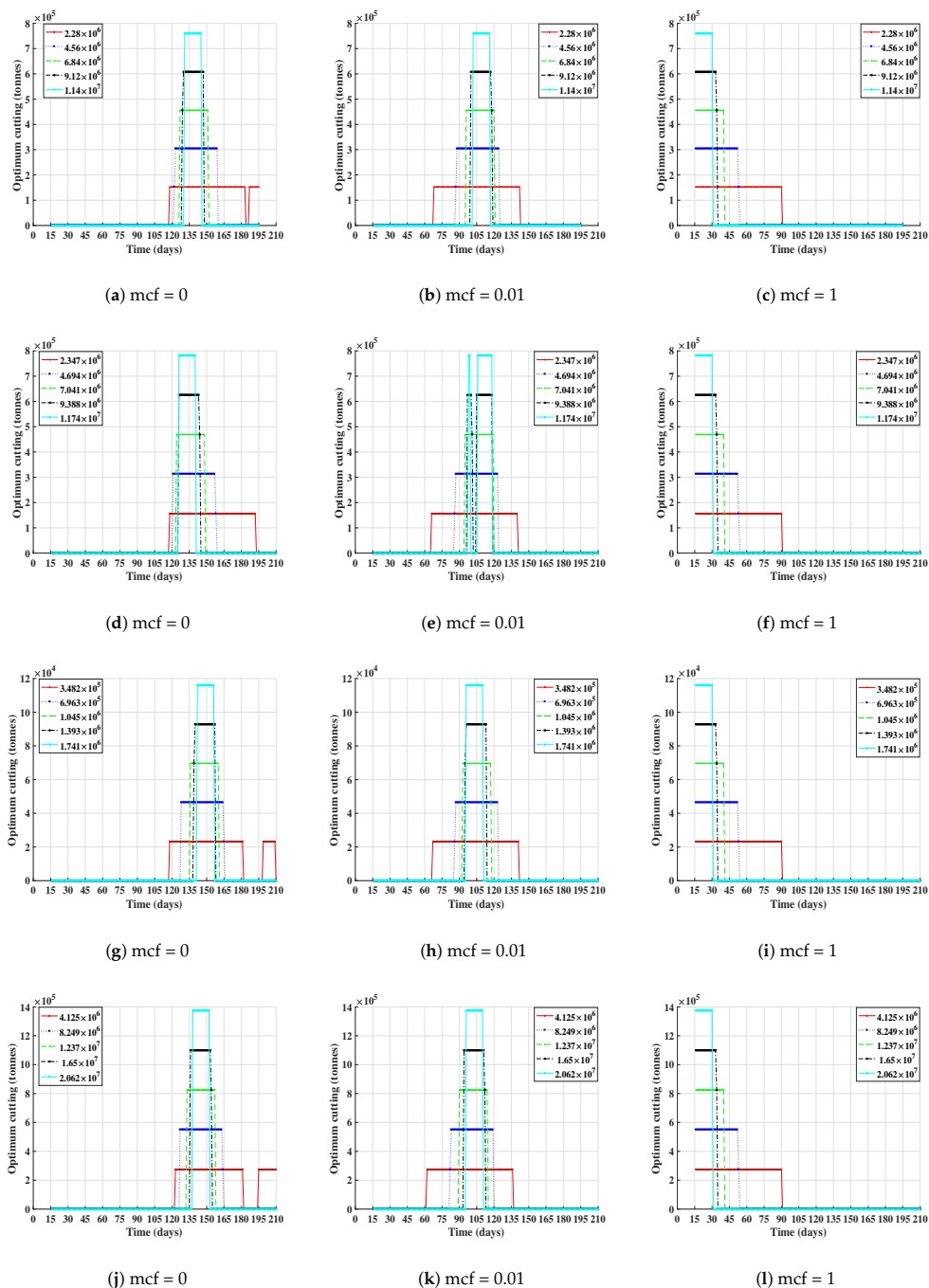

**Figure 1.** Optimum cutting of fresh sugarcane in four regions of Thailand in year 2017/18 showing effects of changing maintenance cost by fraction mcf of actual cost. North: (**a**–**c**), Central: (**d**–**f**), East: (**g**–**i**), North-east: (**j**–**l**).

Some examples of actual cutting patterns obtained from OCSB data [12,13] are shown in Figures 3 and 4 for comparison with the theoretical optimal cutting patterns. It can be seen that the actual cutting patterns for both fresh and fired sugarcane in 2017/18 are similar to the cutting patterns for mcf = 0.01 in Figures 1 and 2 with a maximum cutting of between 0.15–0.2 total available sugarcane in each region. As far as the authors are aware, the actual cutting patterns are due to constraints on cutting on the farms and to the capacities of the sugar mills.

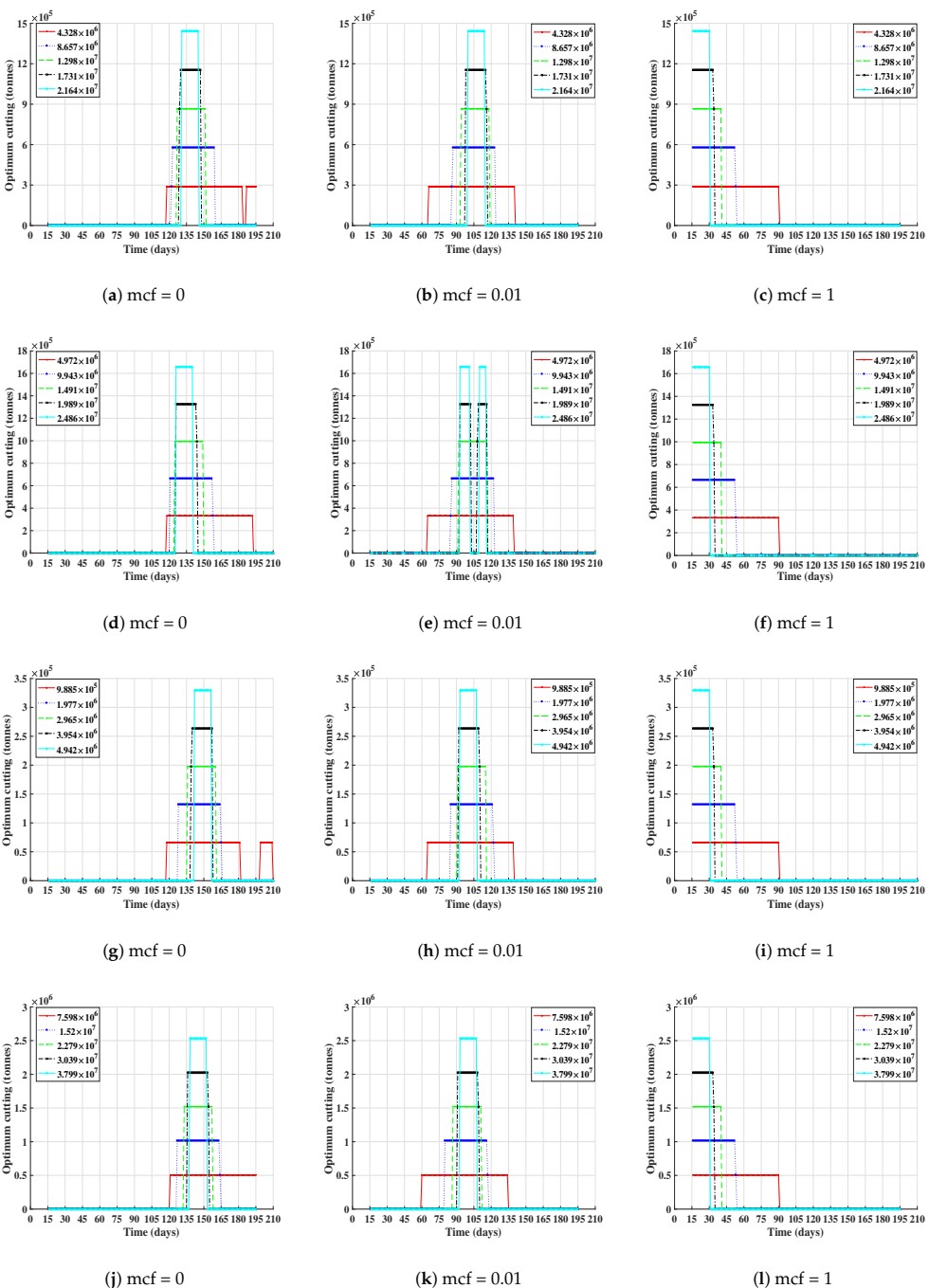

**Figure 2.** Optimum cutting of fired sugarcane in four regions of Thailand in year 2017/18 showing effects of changing maintenance cost by fraction mcf of actual cost. North: (**a**–**c**), Central: (**d**–**f**), East: (**g**–**i**), North-east: (**j**–**l**).

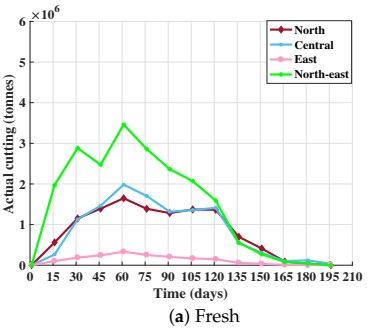 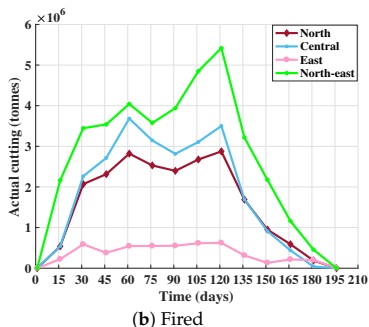

(**a**) Fresh                                   (**b**) Fired

**Figure 3.** Actual cutting patterns for fresh and fired sugarcane for 2017/18.

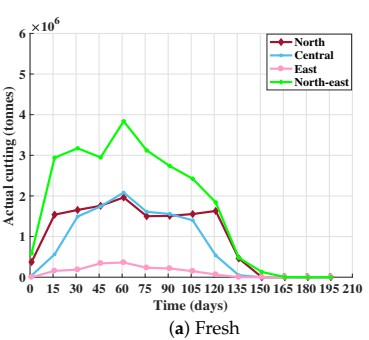 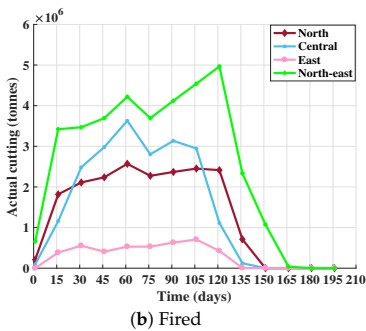

(**a**) Fresh                                   (**b**) Fired

**Figure 4.** Actual cutting patterns for fresh and fired sugarcane for 2018/19.

*5.3. Linear Progamming Dual Variables and Hamiltonian Derivatives*

In the optimal control algorithms in Section 3, we used the maximum values of the Hamiltonian derivatives $\dfrac{\partial H(t_k, x_k, u_k, \lambda_k)}{\partial u_k}$ to find the optimal cutting patterns. It is also well known in linear programming that nonzero values of dual variables $\lambda$ correspond to active constraints and that the values of dual variables correspond to the extra profit (baht) if a cutting constraint can be increased by one unit (tonne). It is therefore expected that maximum Hamiltonian derivative values should correspond to maximum $\lambda$ values. As an example, Figure 5 shows a plot of the Hamiltonian derivative values for fresh sugarcane for the North-east region in 2017/18 and Figure 6 shows a similar plot for the dual variables from linear programming.

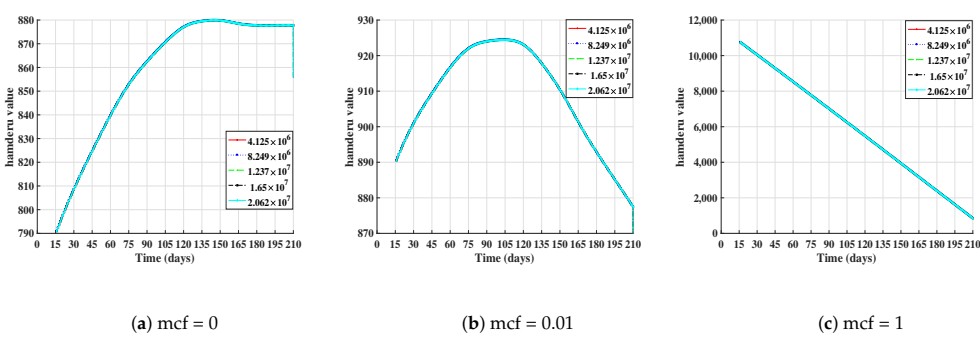

(**a**) mcf = 0              (**b**) mcf = 0.01              (**c**) mcf = 1

**Figure 5.** Example of $\dfrac{\partial H(t_k, x_k, u_k, \lambda_k)}{\partial u_k}$ for fresh sugarcane in North-east region in 2017/18. Compare optimal cutting plots in Figure 1d–f.

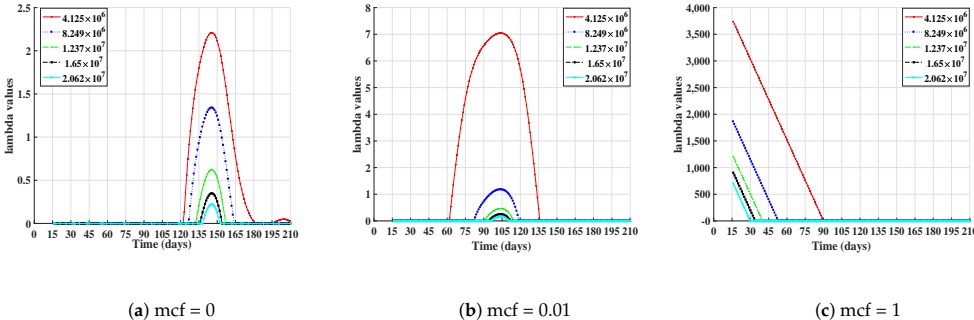

(**a**) mcf = 0          (**b**) mcf = 0.01          (**c**) mcf = 1

**Figure 6.** Example of dual variables from linear programming for fresh sugarcane in North-east region in 2017/18. Compare Hamiltonian derivative plots in Figure 5.

### 5.4. Computation Times

As we have seen, the three optimization methods and the two optimal control methods all give similar optimal cutting patterns and profits. However, there is a big difference in computation times between the different methods. As shown in Table 8, the linear programming, discrete and continuous optimal control methods all give very fast computation times in the order discrete optimal control, linear programming, continuous optimal control. However, the optimal control methods have the advantage that the objective functions and constraints can be nonlinear. As shown in Table 9, the linear programming again gives a fast computation time whereas the bi-objective and quasi-Newton methods are much slower. However, both the bi-objective and quasi-Newton methods again have the advantage that the objective functions and constraints can be nonlinear.

**Table 8.** CPU time (seconds) of fresh and fired sugarcane cutting in each day in crop year 2017/18 and 2018/19.

| Crop Year | Method | mcf | Type of Sugarcane | |
|---|---|---|---|---|
| | | | **Fresh** | **Fired** |
| 2017/18 | discrete | 0 | 0.011 | 0.017 |
| | continuous | | 0.213 | 0.152 |
| | linear programming | | 0.037 | 0.041 |
| | discrete | 1 | 0.010 | 0.013 |
| | continuous | | 0.167 | 0.157 |
| | linear programming | | 0.041 | 0.037 |
| 2018/19 | discrete | 0 | 0.010 | 0.010 |
| | continuous | | 0.171 | 0.162 |
| | linear programming | | 0.031 | 0.034 |
| | discrete | 1 | 0.011 | 0.010 |
| | continuous | | 0.140 | 0.133 |
| | linear programming | | 0.036 | 0.036 |

**Table 9.** CPU time (seconds) of fresh and fired sugarcane cutting in each 15-day period in crop year 2017/18 and 2018/19.

| Crop Year | Method | mcf | Type of Sugarcane | |
|---|---|---|---|---|
| | | | Fresh | Fired |
| **2017/18** | bi-objective | 0 | 0.024 | 0.031 |
| | linear programming | | 0.019 | 0.016 |
| | quasi-Newton | | 83.743 | 72.457 |
| | bi-objective | 1 | 5.226 | 5.385 |
| | linear programming | | 0.038 | 0.031 |
| | quasi-Newton | | 12.905 | 15.426 |
| **2018/19** | bi-objective | 0 | 0.023 | 0.023 |
| | linear programming | | 0.040 | 0.035 |
| | quasi-Newton | | 98.374 | 83.416 |
| | bi-objective | 1 | 5.010 | 4.926 |
| | linear programming | | 0.033 | 0.036 |
| | quasi-Newton | | 13.042 | 15.668 |

## 6. Discussion and Conclusions

We have compared five different optimization and optimal control methods for optimization of sugarcane harvesting in the four sugarcane growing areas of Thailand, namely, North, Central, East and North-east regions for fresh and fired sugarcane for the crop years 2012/13, 2013/14, 2014/15, 2017/18 and 2018/19. To build the mathematical models, we have used price and cost data from the Office of the Cane and Sugar Board [9–13].

For the bi-objective and quasi-Newton methods, we have assumed that there is no growth in sugarcane during the crop year and have divided the crop year into 15-day periods. For the discrete and continuous optimal control methods, we have allowed for the possibility of growth during a crop year through a logistic function and have divided the crop year into 1-day periods. We have also used linear programming to find optimal cutting patterns for both 1-day and 15-day periods for a no-growth model.

As shown in Table 7, the profits obtained from the five different optimization methods have been the same to at least 4-digit accuracy for both fresh and fired for all regions for the crop year 2017/18. We have obtained similar results for both types of sugarcane for all four regions of Thailand for the crop years 2012/13, 2013/14, 2014/15, 2017/18 and 2018/19. It can also be seen from the table that the actual profits are within the range of computed profits for mcf = 1 (full maintenance cost). It is also clear from the results that changes in the maintenance cost are one of the main factors affecting the optimal cutting patterns, with high maintenance costs suggesting cutting as early as possible and low maintenance costs suggesting cutting later.

Although the bi-objective, quasi-Newton, linear programming and optimal control methods give the same optimal cutting patterns and profits for the case of zero growth, there is a big difference in computation times as shown in Tables 8 and 9 with the linear programming and optimal control methods being orders of magnitude faster than the bi-objective and quasi-Newton methods. Further, for the bi-objective, quasi-Newton methods, we have considered 15-day cutting periods for a total of approximately 12 decision variables, whereas for the linear programming and optimal control methods we have considered daily cutting periods for a total of approximately 180 decision variables. However, linear programming can only be used if the growth is zero and the profit function is a linear function of the cutting patterns and the constraints. Finally, we have found that the programming for the discrete optimal control is much simpler than for the continuous optimal control and that the computation times are also shorter.

There are a number of important factors that have not been considered in this paper. For example, there are likely to be changes in the harvesting and transport costs per tonne as a function of amount cut per day and also changes in the costs of harvesting

and transport during the cutting season. Including these effects would make the profit a nonlinear function of control and change the solution from the bang-bang control observed in the models in this paper. Unfortunately, we have not been able to obtain the cost data that would be required to include these effects.

**Author Contributions:** Conceptualization, W.P., S.S., C.K. and E.J.M.; methodology, W.P., S.S., C.K. and E.J.M.; software, W.P., S.S. and E.J.M.; validation, S.S., C.K. and E.J.M.; formal analysis, W.P., S.S. and E.J.M.; writing—original draft preparation, W.P.; writing—review and editing, S.S., C.K. and E.J.M. All authors have read and agreed to the published version of the manuscript.

**Funding:** This research was partially funded by King Mongkut's University of Technology North Bangkok (Contract No. KMUTNB-60-GOV-071) and Centre of Excellence in Mathematics, the Commission on Higher Education, Thailand.

**Institutional Review Board Statement:** Not applicable.

**Informed Consent Statement:** Not applicable.

**Data Availability Statement:** All data on sugarcane in this paper was obtained from the Ministry of Industry and the Ministry of Agriculture and Co-operatives of the Royal Thai government. These data can be found here: [http://www.ocsb.go.th] (accessed on 18 March 2021).

**Acknowledgments:** This research was partially supported by the Graduate College, King Mongkut's University of Technology North Bangkok. The authors would like to express their thanks to the anonymous referees for their time and helpful comments.

**Conflicts of Interest:** The authors declare no conflict of interest.

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
