# Peer review of "An Application of Optimal Control to Sugarcane Harvesting in Thailand"

_computation, doi:10.3390/computation9030036_

Round 1

Reviewer 1 Report

This paper  is thoroughly written and contains a comparative analysis of the outcomes of different standard procedures for solving a good  practical problem. The methods are standard but are used correctly. Only minor technical errors were discovered:

p,2 last line: r is not described correctly.  It has dimensions of 1/ time, and is            the per unit mass rate of change of weight.;

p.4, line 1: add [1]  after "Pontryagin"

Author Response

The authors would like to thank reviewer 1 for their favorable comments on our manuscript.  All changes requested by reviewer 1 have been made as noted below,

Reviewer 1: Comments and Suggestions for Authors

This paper  is thoroughly written and contains a comparative analysis of the outcomes of different standard procedures for solving a good  practical problem. The methods are standard but are used correctly. Only minor technical errors were discovered:

 Comment 1:

p.2 last line: r is not described correctly.  It has dimensions of 1/ time, and is            the per unit mass rate of change of weight.;

Response1:

p.2  We have changed the description of the logistic term in the growth to read the following:

initial time t0 (tonnes) and  rx(t)[1-x(t)/K] - u(t) is a logistic growth function for the rate of increase in weight of sugarcane (tonnes per day) on the farms in a region at time t, where r is a basic growth rate (1/day) and K is a constant which represents the carrying capacity of the farms in the region (tonnes).

Comment 2:

p.4 line 1: add [1]  after "Pontryagin"

Response:2

p.4 line 1:  The citation [1] has been added at the end of the subsection headings of 3.4 and 3.5.

Reviewer 2 Report

I regard the manuscript with ID 1122757 submitted to MDPI Computation as being suitable to publication. It is a profound, designed and well-written work. Its novelty emerges from putting the profit optimization of an agricultural harvesting strategy into the background of control theory. Based on the approximation of optimal control, the authors’ objective is clear as well as their model declaration.

As a critical remark, I would like to warn the authors that the font size of the legends of all the figures is not in accordance with that of the text. (A magnifier, or rather a microscope is necessary to overview the information content of the figures). Thus, because of the trackability of the manuscript outline, this anomaly is to be eliminated before the acceptation.  

Author Response

The authors would like to thank reviewer 2 for their favorable comments on our manuscript.  All changes requested by reviewer 2 have been made as noted below,

Comments and Suggestions for Authors

I regard the manuscript with ID 1122757 submitted to MDPI Computation as being suitable to publication. It is a profound, designed and well-written work. Its novelty emerges from putting the profit optimization of an agricultural harvesting strategy into the background of control theory. Based on the approximation of optimal control, the authors’ objective is clear as well as their model declaration.

Response

The legends to all figures have been increased in size to make them easier to read.